# Operando formation of highly efficient electrocatalysts induced by heteroatom leaching

Cong Liu [1,2], Bingbao Mei[3], Zhaoping Shi[1,2], Zheng Jiang [2,3], Junjie Ge[1], Wei Xing [1,2], Ping Song [1] ✉ & Weilin Xu [1,2] ✉

Heterogeneous nano-electrocatalysts doped with nonmetal atoms have been studied extensively based on the so-called dopant-based active sites, while little attention has been paid to the stability of these dopants under working conditions. In this work, we reveal significantly, when the redox working potential is too low negatively or too high positively, the active sites based on these dopants actually tend to collapse. It means that some previously observed "remarkable catalytic performance" actually originated from some unknown active sites formed in situ. Take the Bi-F for the $CO_2RR$ as an example, results show that the observed remarkable activity and stability were not directly from F-based active sites, but the defective Bi sites formed in situ after the dopant leaching. Such a fact is unveiled from several heteroatom-doped nanocatalysts for four typical reactions ($CO_2RR$, HER, ORR, and OER). This work provides insight into the role of dopants in electrocatalysis.

Nonmetal (N, F, S, etc.)-doped heterogeneous nanocatalysts have been extensively studied for energy-related electrochemical reactions, such as electrochemical $CO_2$ reduction reaction ($CO_2RR$)[1–5], hydrogen evolution reaction (HER)[6,7], oxygen reduction reaction (ORR)[8–10] and oxygen evolution reaction (OER)[11,12]. Nonmetal dopant-based active sites have been taken simply as the main contributors for the high performance of nanocatalysts, due to the boosting activity after doping[13–16]. All the explanations or density functional theory (DFT) calculations about the enhanced catalytic performances were directly based on the heteroatom-based active sites[17–20]. While due to the known limited stability of these nonmetal dopants on supports[21–23], deep understanding to the real roles of these doped heteroatoms under working conditions is highly desirable for the energy-related electrochemical industry[24–28]. While by now little has been done to reveal the working mechanisms of these nonmetal dopants under working conditions[29,30]. For instance, it is still not clear about the potential-dependent stability of dopants during the redox processes, such as $CO_2RR$, HER, ORR, and OER[31–36].

In this work, based on a fluorinated bismuth oxide ($Bi_2O_3$-F) with high $CO_2RR$ performance for the production of formate, besides the expected extremely fast reduction of $Bi_2O_3$-F to metallic Bi-F in seconds, we surprisingly observed the fast leaching of doped F from Bi support in minutes to form defective Bi in situ. It means that the observed high activity and stability of $CO_2RR$ performance of "Bi-F" doesn't originate directly from the F-based active sites, but the defective Bi sites formed in-situ after the leaching of F. Such fact was further confirmed by DFT calculations. The fast leaching of heteroatom-dopants was further observed on F, N-doped carbon surfaces during $CO_2RR$/HER/OER processes. While as for the ORR process on F, N-doped carbon surfaces, it shows that the F- and N-dopants are very stable in the typical ORR potential window, which is not too far from the relative hydrogen electrode (RHE). All these results indicate that the observed high performances of the nonmetal heteroatoms-doped electrocatalysts for the redox process at too high positive or too low negative potentials are usually not from the dopants directly but from the sites formed in-situ after the leaching of these

[1]State Key Laboratory of Electroanalytical Chemistry, & Jilin Province Key Laboratory of Low Carbon Chemical Power, Changchun Institute of Applied Chemistry, Chinese Academy of Sciences, Changchun 130022, China. [2]School of Applied Chemistry and Engineering, University of Science and Technology of China, Hefei 230026, China. [3]Shanghai Synchrotron Radiation Facility, Shanghai Institute of Applied Physics, Chinese Academy of Sciences, Shanghai 201204, China. ✉e-mail: songping@ciac.ac.cn; weilinxu@ciac.ac.cn

heteroatoms. It means that the conclusions made in previous work especially that for $CO_2RR$, HER, and OER need to be reconsidered. This work provides insight into the real role of heteroatoms doped and the real activity origin of nonmetal-doped materials under working conditions and offers a protocol to engineer highly efficient active sites with respect to dopant leaching process.

## Results and discussion

We firstly prepared pure bismuth oxide nanoparticles ($Bi_2O_3$) and fluorinated bismuth oxide nanoparticles ($Bi_2O_3$-F) via the air annealing process based on the precursor bismuth nanoparticles (Pre Bi) and NaF (see details in the Methods). The scanning electron microscopy (SEM) images for $Bi_2O_3$ and $Bi_2O_3$-F indicate that the air annealing does not change the morphology of Pre Bi (Fig. 1a, c and Supplementary Fig. 1). The high-resolution transmission electron microscope (HRTEM) analysis indicates that the lattice spacing of (221)-facet of $Bi_2O_3$-F is slightly larger than that of $Bi_2O_3$ (Fig. 1b, d), consistent with the X-ray powder diffraction (XRD) results shown in Supplementary Fig. 1. Such facts indicate that the F-doping can tune the lattice structure of $Bi_2O_3$ via the formation of Bi-F bonding (Supplementary Fig. 2, Supplementary Table 1)[37]. The $CO_2RR$ performance of both $Bi_2O_3$ and $Bi_2O_3$-F were evaluated by determining the Faradaic efficiency (FE) of products with

online-connected gas chromatograph and $^1$H-NMR spectra (Supplementary Figs. 3 and 4). As shown in Fig. 1e, f, compared with $Bi_2O_3$, $Bi_2O_3$-F presents much higher Faradaic efficiencies for the $HCOO^-$ production ($FE_{HCOO^-}$) in a wide potential range (−0.77 V to −1.27 V vs. RHE). Partial current densities for $HCOO^-$ ($j_{HCOO^-}$) confirm the higher catalytic performance of $Bi_2O_3$-F than $Bi_2O_3$ (Fig. 1g, Supplementary Fig. 5, Supplementary Fig. 6, and Supplementary Table 2). We further studied the $CO_2RR$ stabilities of catalytic performance of both $Bi_2O_3$ and $Bi_2O_3$-F (Supplementary Fig. 7). As shown in Fig. 1h, after a long-term (100 hours) continuous $CO_2RR$ process, the $FE_{HCOO^-}$ on $Bi_2O_3$-F decreased only 11%, while the $FE_{HCOO^-}$ on $Bi_2O_3$ decreased up to 22%, indicating a much higher stability of $Bi_2O_3$-F than $Bi_2O_3$ for the formate production from $CO_2RR$. Such improvement obviously originates from the doping of F. To here, All these results indicate that the F doping indeed can improve the $CO_2RR$ catalytic performance of $Bi_2O_3$, consistent with previous observations about the performance enhancement of electrocatalysts by the doping of non-metal atoms on supports[35].

To further understand the role of F or catalytic mechanism of F-based active sites, we first investigate the possible component variation of the catalyst during the $CO_2RR$ process. Operando Raman spectra (Fig. 2a and Supplementary Fig. 8) show firmly that the $Bi_2O_3$

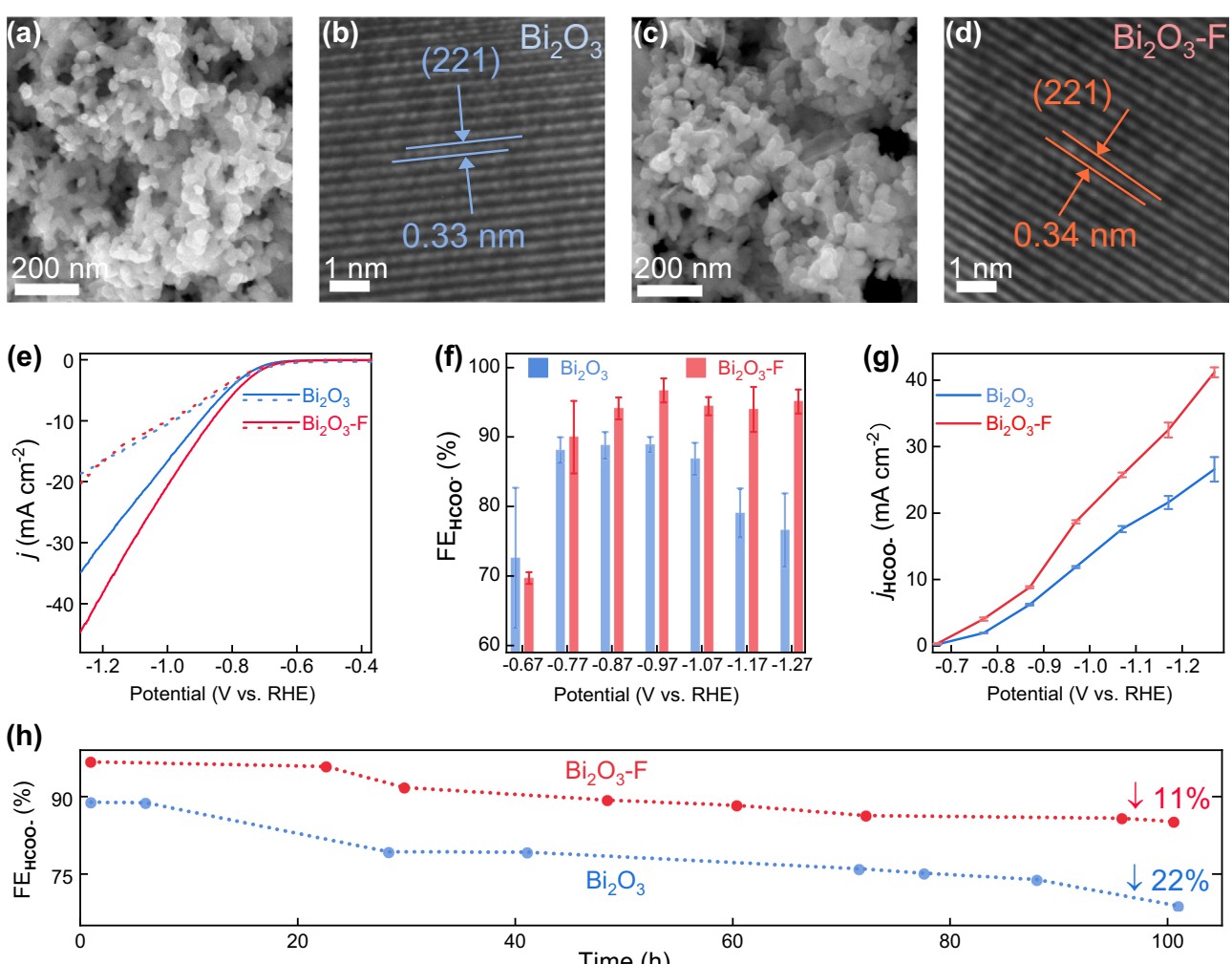

**Fig. 1 | Structural and electrochemical properties. a** SEM and **b** HRTEM images of $Bi_2O_3$. **c** SEM and **d** HRTEM images of $Bi_2O_3$-F. Lattice spacing is represented by the blue and red mark. **a**, **c** Scale bar = 200 nm; **b**, **d** Scale bar = 1 nm. **e** Comparison of linear sweep voltammetric (LSV) with pH corrections for $CO_2$ (solid line, pH 7.2) and Ar (dash line, pH 8.8) saturated electrolytes. Colors in blue and red represent $Bi_2O_3$ and $Bi_2O_3$-F, respectively. **f** Comparison of $FE_{HCOO^-}$ at different applied potentials ranging from −0.67 V to −1.27 V (RHE). Error bars correspond to the standard deviation of three independent measurements. **g** Comparison of $j_{HCOO^-}$ at different applied potentials ranging from −0.67 V to −1.27 V (RHE). **h** Long-term durability of formate selectivity of the $Bi_2O_3$ (blue), and $Bi_2O_3$-F (red) under chronoamperometry test (−0.97 V vs. RHE, 0.5 M $KHCO_3$).

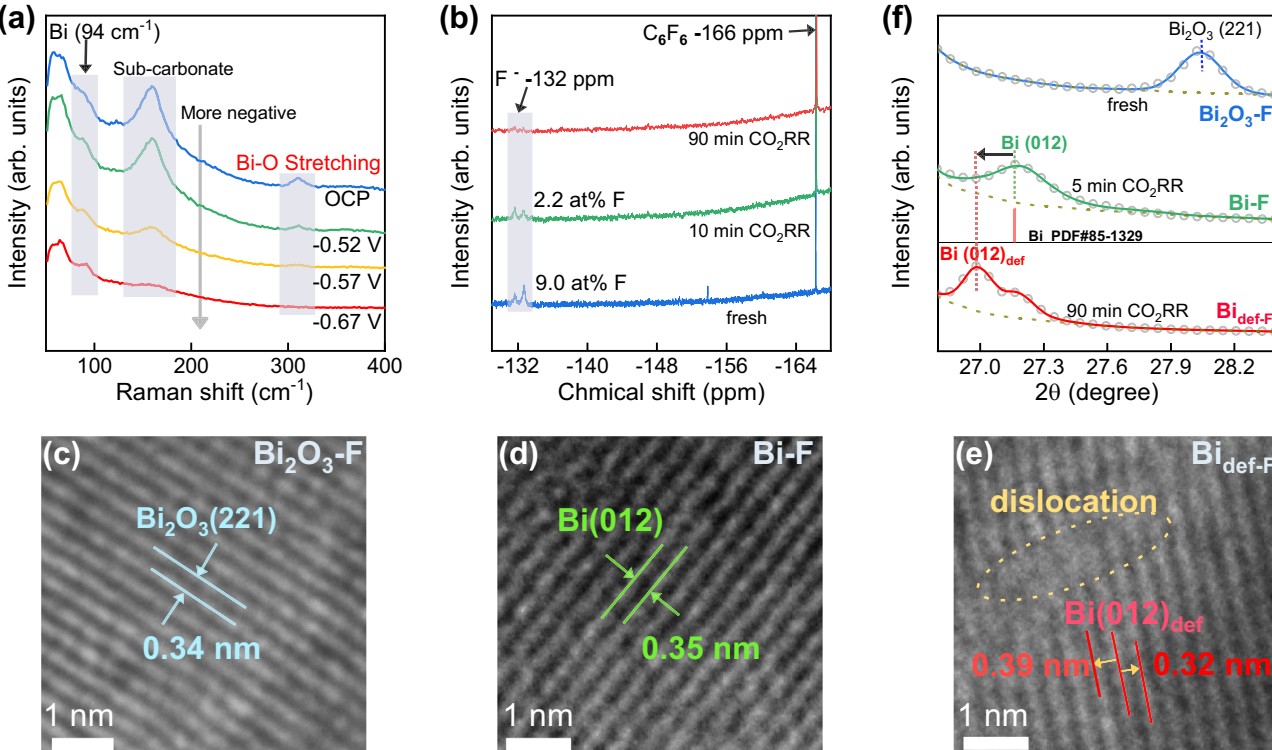

**Fig. 2 | Structural evolution of Bi$_2$O$_3$-F during CO$_2$RR. a** Operando Raman spectra of Bi$_2$O$_3$-F during CO$_2$RR process in the potential range from −0.52 V to −0.67 V vs. RHE in CO$_2$ saturated 0.5 M KHCO$_3$. The sweep duration for each potential is 30 seconds. **b** Time-dependent $^{19}$F-NMR spectra of Bi$_2$O$_3$-F after CO$_2$RR at -0.97 V. HRTEM images of **c** fresh Bi$_2$O$_3$-F, **d** Bi-F and **e** Bi$_{def-F}$ (note: Bi-F and Bi$_{def-F}$ are derived from the fresh Bi$_2$O$_3$-F after 5 min and 90 min CO$_2$RR, respectively). Scale bar = 1 nm. **f** The Rietveld-refined XRD results of Bi$_2$O$_3$-F, Bi-F, and Bi$_{def-F}$. Colors in blue, green, and red represent Bi$_2$O$_3$-F, Bi-F, and Bi$_2$O$_3$-F, respectively.

can be reduced fast to metallic Bi in seconds under the potential for CO$_2$RR as indicated by the fast disappearance of Bi-O stretch peaks (312 cm$^{-1}$) and the appearance of a peak for metallic Bi (94 cm$^{-1}$)[38,39]. Such fact means that the real components for long-term CO$_2$RR process observed above is from Bi or Bi-F rather than the original Bi$_2$O$_3$ or Bi$_2$O$_3$-F. To further unveil the status of dopants in Bi$_2$O$_3$-F during the CO$_2$RR process, the fluoride species on Bi$_2$O$_3$-F were investigated by $^{19}$F-NMR spectra after certain time of CO$_2$RR at -0.97 V (Supplementary Fig. 9). Surprisingly, Fig. 2b shows, after 10 minutes of CO$_2$RR, the fluorine content on Bi$_2$O$_3$-F dropped hugely from 9.0at% to 2.2at% and the fluorine almost disappears after 90 minutes. The above facts mean that two sequential reconstructions occur quickly on Bi$_2$O$_3$-F during the CO$_2$RR process: the first is the fast reduction from Bi$_2$O$_3$-F to F-doped metal (Bi-F) in seconds, the second is the leaching of F to produce defective Bi surface (Bi$_{def-F}$) in minutes. So, the long-term CO$_2$RR actually mainly occurs on Bi$_{def-F}$. From the performance shown in Fig. 1, one can tell that the defective Bi surface (Bi$_{def-F}$) possesses much higher CO$_2$RR activity and stability than pure smooth Bi surface.

We next conducted the lattice analysis on the above three catalysts (including fresh Bi$_2$O$_3$-F) to gain more insights. The analyzes of the HRTEM (Fig. 2c–e) show that the reduction of Bi$_2$O$_3$-F can increase slightly the lattice distance (from 0.34 nm to 0.35 nm) and the further leaching of F enables the local dislocation as indicated by the simultaneous appearance of both larger (from 0.35 nm to 0.39 nm) and smaller (from 0.35 nm to 0.32 nm) lattice distances. Such a unique defect feature observed on Bi$_{def-F}$ surface cannot be found on the surface of reduced Bi$_2$O$_3$ (Supplementary Fig. 10), confirming such unique feature originates in situ from the F leaching. The XRD patterns (Fig. 2f, Supplementary Fig. 11, and Supplementary Table 3) further unveil a new facet of defective Bi(012) formed-in situ after the F leaching as indicated by the peak at 27.0 degrees observed on Bi$_{def-F}$, indicating that the Bi$_{def-F}$ undergoes lattice expansion after the F

leaching[40]. To here, one can conclude that the leaching of F atoms from Bi surface induces the local dislocation and then the in-situ formation of defective sites by removing some Bi atoms along with.

We further did the density functional theory (DFT) calculation to understand the activity origin of Bi-F and Bi$_{def-F}$ for CO$_2$RR. Firstly, based on the above experimental results, we constructed the metallic Bi (hexagonal, Bi(012)), one fluorine atom modified metallic Bi (Bi(012)-F), and defective Bi(012) (Bi(012)$_{def-F}$, in-situ formed defective sites after F-leaching from Bi(012)-F). Optimized geometric structures can be found in Supplementary Fig. 12. Notably, after the F-leaching, the formed defective sites enable the change of geometric structures of the neighbor Bi atom (Fig. 3a), which is in good agreement with the outcomes in Fig. 2c–e. To gain more insights into these three catalytic surfaces, the electron localization function (ELF) was calculated to measure the degree of electronic localization (the probability of finding an same spin electron in the nearby space)[41]. As shown in Fig. 3b, the Bi(012)$_{def-F}$ displays higher electron delocalization around the formed defective sites than that on Bi(012) and Bi(012)-F. Correspondingly, the detailed projected density of states (PDOS) reveals that the electron density near the Fermi level (E$_f$) became higher in Bi(012)$_{def-F}$ surface (Supplementary Fig. 13), which could give rise to the more efficient surface electron transfer[42]. The above results confirm that the electronic structure of Bi(012)$_{def-F}$ surface was changed after defective sites formed due to the leaching, which can further affect positively the CO$_2$RR catalytic activity.

For the formate production on these Bi-based catalysts, the formation of *OCHO intermediate was considered as the primary pathway[43–45]. Therefore, we calculated the adsorption energies of *OCHO on these three catalytic surfaces: Bi(012), Bi(012)-F, and Bi(012)$_{def-F}$ (Supplementary Fig. 14). DFT calculations show that the adsorption energies of *OCHO are −1.3, −1.2, and −1.9 eV on them, respectively (Fig. 3c). We further calculated the Gibbs free energy (ΔG)

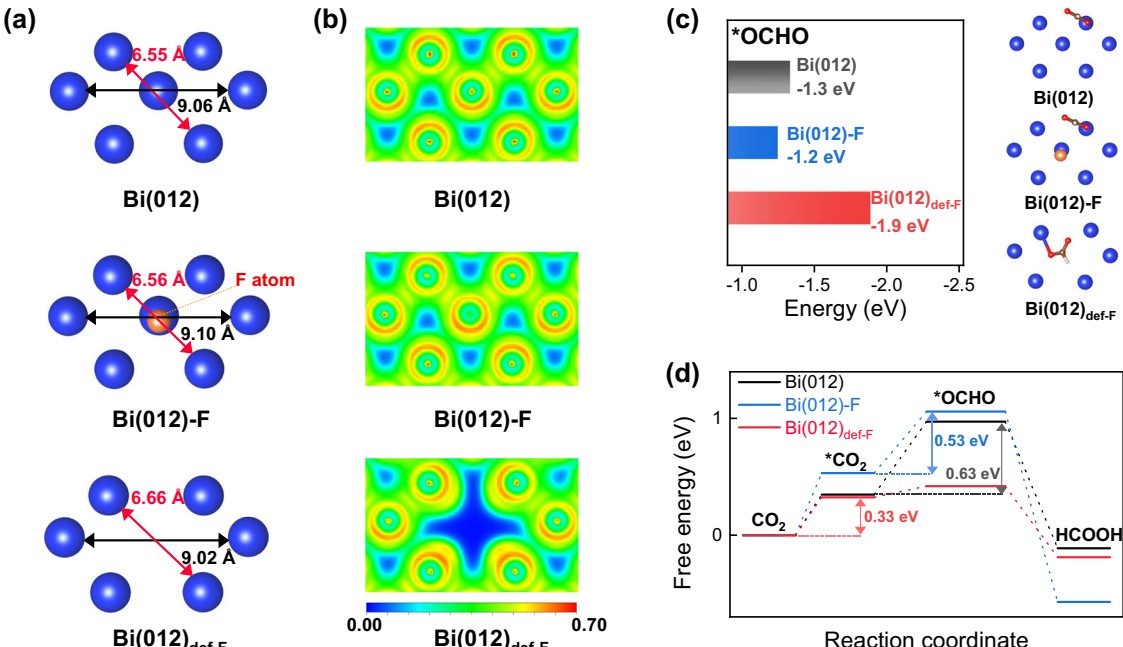

**Fig. 3 | Theoretical calculations. a** The partial slice of optimized geometric structures (Bi(012), Bi(012)-F, and Bi(012)$_{def-F}$). The introduction of a dopant atom (F, orange color) enables the change of distance between the neighbor Bi atoms. **b** Comparison of electron localization function calculated from the optimized geometric structures. **c** Comparison of the binding energy of *OCHO in the different catalytic surfaces. Colors in grey, blue, and red represent Bi(012), Bi(012)-F, and Bi(012)$_{def-F}$, respectively. Right: The corresponding adsorption structures of *OCHO. **d** Free energy profiles for the formation of *OCHO intermediate on the catalytic surface of Bi(012), Bi(012)-F, and Bi(012)$_{def-F}$.

for the *OCHO formation (0.63, 0.53, and 0.09 eV respectively, Fig. 3d). These results reveal that the defective sites facilitate the formation of *OCHO. More importantly, after the leaching of dopant atoms, the reaction barrier of the rate-limiting step in the Bi$_{def-F}$ surface further decreases to 0.33 eV (lower than Bi(012) and Bi(012)-F), indicating that the observed much higher $CO_2RR$ performance of $Bi_2O_3$-F (Fig. 1) should be attributed to the defective sites formed in situ after the F leaching. We further calculated the charge density difference and the electron transfer (bader charge) on these three intermediate-adsorbed catalytic surfaces: Bi(012), Bi(012)-F, and Bi(012)$_{def-F}$. As shown in Supplementary Fig. 15, the F doping can weaken the electron transfer from Bi to O and then weaken the adsorption of the intermediate on Bi, while after the formation of defects via the leaching of F, the electron transfer from Bi to O can be enhanced, which then enhances the adsorption of the intermediate on Bi.

In addition, based on the experimental results shown in Fig. 2e, f about the F-leaching induced lattice expansion, we further studied its effect on the $CO_2RR$ performance of Bi. The results (Supplementary Fig. 16 and Supplementary Note 1) showed that such lattice expansion can improve slightly its catalytic activity for $CO_2RR$ via a rate-limiting step of $CO_2$ activation.

Since heterogeneous nano-electrocatalysts doped with nonmetal atoms have been studied extensively for all kinds of redox reactions in a wide potential range[46–48], it is very necessary to further confirm the universality of above unveiled in-situ formation of active sites via the leaching of dopants on heterogeneous nanocatalysts. For this goal, we fabricated F- and N- doped carbon black based on BLACK PEARLS (BP) (BP-F and BP-N, Supplementary Fig. 17 and Supplementary Fig. 18) to gain more insights into the stability of typical dopants of F and N in a much wider potential range. Based on the standard electrode potentials of half-electrochemical thermodynamic reaction (Supplementary Table 4), we choose to study the four most typical and important energy chemistry processes ($CO_2RR$, HER, ORR, and OER) covering the electrode potential from −1.0 V to 1.8 V vs. RHE. As shown in Fig. 4a, with potential lower than −0.5 V in $CO_2$-saturated 0.5 M $KHCO_3$, BP-F

and BP-N exhibit catalytic activity for $CO_2RR$ to produce CO (Supplementary Fig. 19 and Supplementary Fig. 20). In the potential range between −0.7 V and 1.0 V, BP-F and BP-N exhibited remarkable HER and ORR activities (Fig. 4b). At much higher positive potentials ( > 1.0 V), BP-F and BP-N also present OER activities. Based on such facts, BP-F and BP-N are taken as model catalysts to validate the dopant stability under different test intervals.

The typical ex-situ XPS analysis (Fig. 4c, d) shows that the F- and N-dopants were rapidly removed at negative reduction potential when during the $CO_2RR$ (-0.87 V vs. RHE) while still maintained steadily when during the ORR at 0 V vs. RHE. Based on such qualitative analysis, the leaching processes of both F and N doped on carbon were analyzed deeply for the catalytic processes of $CO_2RR$, HER, ORR, and OER, respectively. As shown in Fig. 4e, g, at more negative potentials such as -0.87 V and −0.67 V for $CO_2RR$, the F- and N-dopants leached rapidly, while the sustained high catalytic activities of these two catalysts in a long-time testing window (Supplementary Fig. 21) indicate that the observed long-term activities originate from defects formed after the dopant leaching (Supplementary Figs. 22 and 23). With the potential increase up to -0.37 V and -0.47 V for HER (Fig. 4e, g), the leaching rates of both F- and N-dopants decrease a little bit. Moreover, the catalytic activity of HER in BP-F increased after the dopant leaching, suggesting the long-term HER activity of BP-F is also from the active sites formed in situ (Supplementary Fig. 24). With the potential further increase up to 0.0 V, 0.3 V, and 0.6 V for ORR (Fig. 4e, g), the heteroatoms of F and N were stable in the potential window for ORR, implying that the doped heteroatoms on support indeed are the main contributors for the long-term ORR process (Supplementary Fig. 25). While at much higher positive potentials for such as OER (1.5 V and 1.8 V), the leaching occurs fast again (Fig. 4e, g). Figure 4f, h summarizes the leaching rates of dopants (F and N) at different test intervals, clearly elucidating the potential dependent leaching of dopants.

To further validate whether such leaching is related to the catalytic reaction or not, we conducted control experiments with these two catalysts in Ar-saturated 0.5 M $KHCO_3$ at -0.87 V. Supplementary

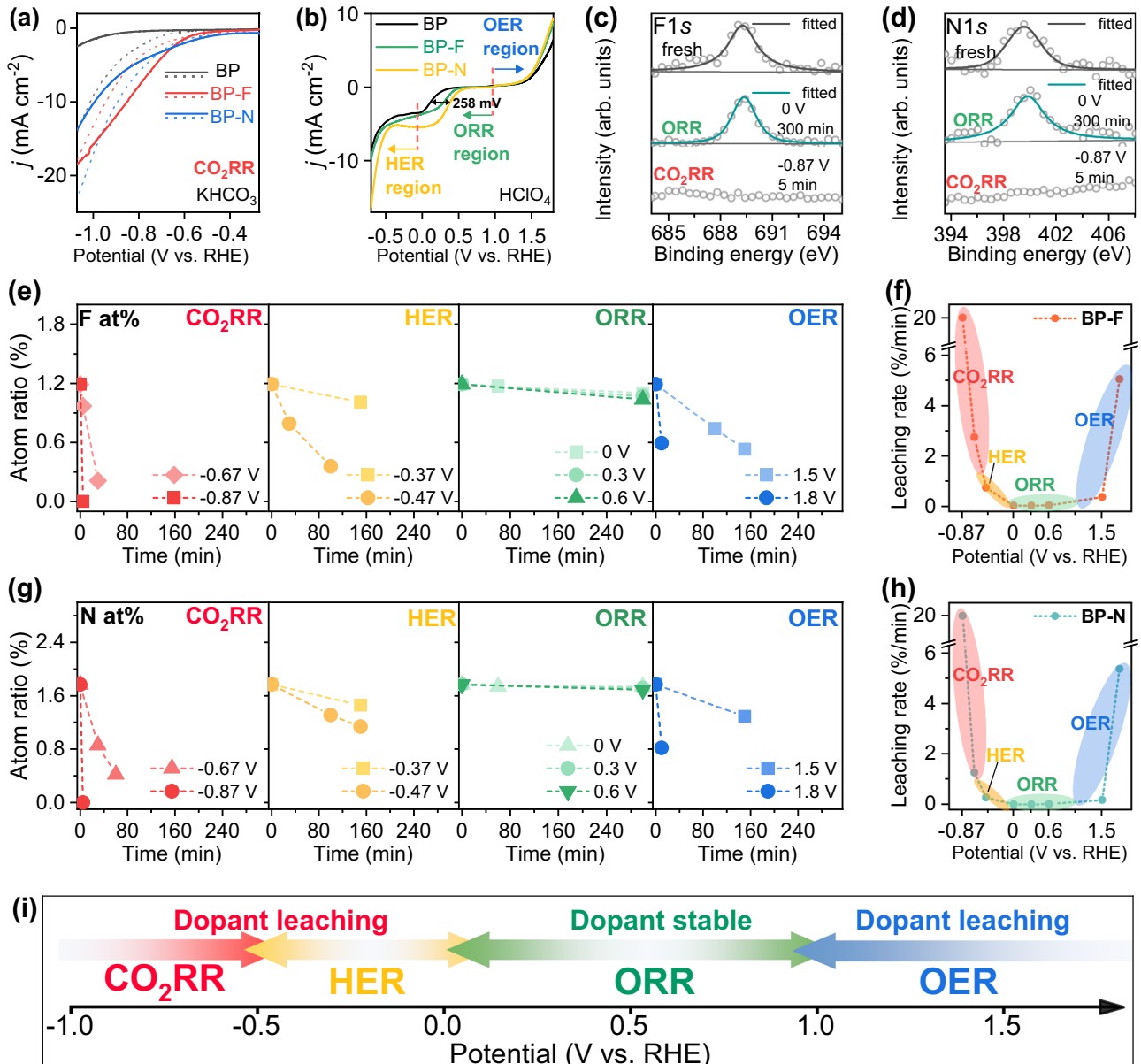

**Fig. 4 | Potential-driven dopant leaching in different materials. a** Comparison of LSV with pH corrections for $CO_2$ (solid line, pH 7.2) and Ar (dash line, pH 8.8) saturated electrolytes (0.5 M $KHCO_3$). Colors in black, red, and blue represent BP, BP-F, and BP-N, respectively. **b** The linear sweep curves of the catalyst were tested in $O_2$-saturated 0.1 M $HClO_4$ (rotation rate 1600 rpm, scan rate of 5 mV $s^{-1}$). The test interval is divided into three range areas: OER, ORR, and HER regions. Colors in black, green, and yellow represent BP, BP-F, and BP-N, respectively. **c** F1$s$ XPS of BP-F and **d** N1$s$ XPS of BP-N after 5 min $CO_2RR$ (-0.87 V vs. RHE) and 300 min ORR (0 V vs. RHE). **e** F-dopant content and (**g**) N-dopant content as a function of potential applied time in the $CO_2RR$, HER, ORR, and OER. The ORR and OER were tested in $O_2$-saturated 0.1 M $HClO_4$, HER was tested in Ar-saturated 0.1 M $HClO_4$, and $CO_2RR$ was tested in $CO_2$-saturated 0.5 M $KHCO_3$. **f** F-dopant and **h** N-dopant leaching rates as a function of potential applied. The leaching rates of dopants in the specific voltage (unit: %/min) is based on the leaching degree divided by the leaching time. **i** The schematic diagram of dopant leaching (BP-F and BP-N) in the corresponding electrochemical processes.

Fig. 26 shows clearly that the leaching rates of both F and N are the same as that observed during the $CO_2RR$ process at the same potential, indicating that the leaching is mainly driven by the electrode potential rather than the redox reaction. We studied the dopant-dependent leaching rates at the same potential (Supplementary Fig. 27) and found that the leaching of F is slightly faster than N from carbon surface probably due to the different binding structures of C-F and C-N[49,50]. Considering the different electrochemical reaction conditions that have appeared in previous studies[35,51–54], we also compared the dopant leaching in the different type of electrolytes (Supplementary Figs. 28 and 29), and different types of host material (Supplementary Fig. 30). Based on the above control experiments, we further conclude that the

dopant leaching rates are related to the type of electrolyte, dopant atom, or host material, but the leaching mechanism is same: the driving force of leaching is the voltage.

To further confirm the leaching window and stable window of dopants N and F from a theoretical perspective, we calculated the surface Pourbaix diagram to unfold the operando surface condition. As shown in Supplementary Fig. 31[55–57], the obtained leaching Pourbaix diagram reveals that the F- and N-dopants leach at either too high or too low potential ranges while a relatively stable potential interval exists in the middle range of the voltage, which is in good agreement with the experimental outcomes (Fig. 4). Based on the above facts, the stabilities of dopant atoms (mainly F, N studied here) in the different

electrochemical processes were summarized in Fig. 4i. The potential-induced leaching mechanism of dopants (F and N) or in-situ formation of active sites via the leaching of dopants revealed here inspire us to rethink the design and the activity origin of dopant-induced highly efficient nanocatalysts.

In summary, we studied the stability of dopants on heterogeneous nano electrocatalysts for the four most representative and important redox reactions under working conditions. It reveals significantly that the active sites based on these dopants actually can collapse due to the fast leaching of dopants when the redox working potential is too low or too high. It means that some previously observed "remarkable catalytic activity and stability" actually originated from the active sites formed in situ. Such a fact is unveiled from several heteroatom-doped nanocatalysts (Bi-F, C-F, and C-N) for CO$_2$RR, HER, ORR, and OER, respectively. Take the Bi-F for the electrocatalytic CO$_2$RR to produce formate as an example, results show that the observed remarkable activity and stability is not directly from F-based active sites, but the defective Bi sites formed in situ after the fast leaching of F at negative potentials. Similar results are observed on F, N-doped carbon (BP-F and BP-N) in the potential window for CO$_2$RR and HER, respectively. While during the ORR process on BP-F and BP-N, the dopants are stable. At more positive potentials such as for OER process, the fast leaching of dopants occurs again. The work provides insight into the real role of heteroatoms doped on nanocomposites for electrocatalysis and a protocol for the in-situ formation of highly efficient active sites on functional materials via the leaching of dopants.

## Methods

### Materials

Bismuth (III) chloride was purchased from adamas-beta, China; Polyethene glycol (Mn 1000), hydrochloric acid (HCl, 30 wt%) and ethanol were purchased from Xilong scientific, China; 2-ethoxyethanol, sodium fluoride (NaF), ammonium fluoride (NH$_4$F), melamine (C$_3$H$_6$N$_6$), tetradecylamine (TDA), tin tetrachloride (SnCl$_4$), potassium bicarbonate (KHCO$_3$), Deuterium oxide (D$_2$O, 99.9%) and hexafluorobenzene (C$_6$F$_6$, 99.9%) were purchased from Aladdin, China; Dimethyl sulfoxide (DMSO, 99.99%) and Nafion solutions (5 wt%) were purchased from Sigma-Aldrich. Carbon black (BLACK PEARLS 2000, noted as BP) was purchased from Cabot, America. All chemicals were used without further purification. The resistivity of deionized water was 18.2 MΩ cm in solution preparations.

### Preparation of precursor bismuth nanoparticles (Pre Bi)

Bismuth (III) chloride (100 mg), polyethene glycol (1.2 g) were dissolved into 60 mL 2-ethoxyethanol, then the sample was under ultrasonication to form a uniform and transparent solution. The solution of NaBH$_4$ (10 mL, 60 mmol) was used to reduce the bismuth (III) chloride. After twice washing with water and ethanol the mixture was collected by filtration. Finally, the sample was dried in a vacuum oven at 50 °C and then collected for further experiments.

### Preparation of bismuth oxide nanoparticles (Bi$_2$O$_3$, Bi$_2$O$_3$-F)

Pre Bi (10 mg) was mixed with a certain amount of NaF (0 mg and 1.6 mg) and annealed at 200 °C in the air for 6 h to obtain bismuth oxide or fluorinated bismuth oxide nanoparticles (Bi$_2$O$_3$ and Bi$_2$O$_3$-F). After washing with enough water, the sample was dried in a vacuum oven at 50 °C and then collected for further experiments.

### Preparation of F-doped carbon (BP-F)

Carbon black (BP) was mixed with a certain amount NH$_4$F (30 wt%), then the mixture was dispersed in 50% ethanol solution. After fiercely stirring, the mixture was dried under a vacuum at 50 °C and then pyrolyzed at 400 °C under argon atmosphere for 6 h with a flow rate of 80 mL min$^{-1}$. After washing with enough water, the sample was dried in a vacuum oven at 50 °C and then collected for further experiments.

### Preparation of N-doped carbon (BP-N)

Carbon black (BP) was mixed with a certain amount of melamine (mass ratio 1:15). After repeated grinding, the mixture was pyrolyzed at 900 °C under argon atmosphere for 1 h with a flow rate of 80 mL min$^{-1}$. After washing with enough water, the sample was dried in a vacuum oven at 50 °C and then collected for further experiments.

### Preparation of fluorinated tin oxide (SnO$_2$-F)

The synthesis method was derived from ref. 35,58. SnCl$_4$ (2.61 g) and NH$_4$F (0.374 g) were mixed in TDA solution (1.28 g TDA, 85 mL ethanol, and 160 mL deionized water). The ammonium hydroxide solution (1.5 mmol L$^{-1}$, 200 mL) was added dropwise, followed by stirring for 1 h. Then the suspension was refluxed at 80 °C for 72 h. After cooling to room temperature and ethanol washing, the mixture was hydrothermally treated at 120 °C for 24 h. After the hydrothermal process, the sample was washed with adequate ethanol and dried in a freeze dryer. Finally, the drying sample was calcined at 400 °C in the air for 3 h and then collected for further experiments.

### Structural characterizations

The morphology was characterized by scanning electron microscopy (SEM, ZEISS Sigma-300) and transmission electron microscopy (TEM, JEOL JEM-2100, 200 kV). The basic physical structure was characterized by X-ray diffraction (XRD, Rigaku-D/MAX-PC 2500, Cu Kα source), confocal Raman spectroscopy (Horiba-JY Labram-010, 532 nm Nd laser), and X-ray photoelectron spectroscopy (XPS, Thermo ESCALAB 250, Al Kα source). $^1$H-NMR and $^{19}$F-NMR were performed on a BRUKER ADVANCE-III 500HD (Switzerland). X-ray absorption spectras (XAS) were performed on the BL14W1 beamline at the Shanghai Synchrotron Radiation Facility (SSRF). Electrochemical data were collected by electrochemical work station (CH Instruments, CHI 760E).

### Electrochemical measurements in a three-electrode cell

The HER, ORR, and OER performance was tested by Pine Modulated Speed Rotator (PINE, America). 1 mg samples, 10 μL Nafion solutions (5 wt%), and 200 μL ethanol were mixed and dispersed by ultrasonication to form catalyst ink. 10 μL ink was evenly drop-cast onto the surface of the rotating disk electrode (RDE) to work as working electrode. A carbon rod and Ag/AgCl reference electrode (3.5 M KCl aqueous used as the filling solution) were used as the counter and reference electrode, respectively. The HER, ORR, and OER performance were determined by linear sweep curves test (LSV) in Ar/O$_2$ saturated electrolytes (scan rate: 5 mV s$^{-1}$).

### Electrochemical CO$_2$ reduction

The detection of typical CO$_2$RR product was performed with an H-type two-compartment electrochemical cell (H-cell). Cation-exchange membrane (Nafion 117, Dupant Company) was used to separate working and counter electrodes (cathodic and anodic compartments). 210 μL ink (1 mg catalysts) was evenly drop-cast onto the surface of carbon paper (1×1 cm$^2$) as a working electrode. Electrode potentials (Ag/AgCl) were converted to potentials versus the reversible hydrogen electrode (RHE) by:

$$E_{RHE} = E_{Ag/AgCl} + 0.0591 \times pH + 0.205 \qquad (1)$$

Before the experiments, the electrolyte (50 ml 0.5 M KHCO$_3$) was saturated with CO$_2$ (50 mL min$^{-1}$) at least 30 min at room temperature and ambient pressure. Gas chromatograph (GC, Thermo Trace 1300) was online-connected with the H-cell, with a Molecular Sieve 5 A capillary column and a packed Carboxen-10000 column. Helium (99.999%, Juyang Co. Ltd.) was used as the GC carrier gas. Hydrogen and carbon monoxide were quantified by thermal conductivity and flame ionization detector (TCD, FID, equipped with a methanizer), respectively. The Faradaic efficiency (FE) and the partial current

densities of gas products ($H_2$,CO) were calculated as below:

$$FE_S = \frac{2F\upsilon_s GP_0}{RT_0 i_{total}} \times 100\% \qquad (2)$$

$$j_{H_2} = \frac{FE_{H_2} \times i_{total}}{\text{electrode area}} \qquad (3)$$

$$j_{CO} = \frac{FE_{CO} \times i_{total}}{\text{electrode area}} \qquad (4)$$

Where $V_s$ represents hydrogen and carbon monoxide volume concentrations from the exhaust gas of H-cell (GC quantified), $P_0 = 1.013$ bar and $T_0 = 298.15$ K, G represents gas flow rate (mL min$^{-1}$, exit of cathodic compartment), $i_{total}$ represents steady-state cell current (mA), F = 96485 C mol$^{-1}$, R = 8.314 J mol$^{-1}$ K$^{-1}$.

Liquid product was quantified by $^1$H-NMR spectra. Typical NMR samples were mixed by 500 μL electrolyte, 100 μL $D_2O$, and 0.05 μL DMSO (internal standard). The FE can be calculated as follows:

$$FE_{HCOO^-} = \frac{2F \times n_{HCOO^-}}{i_{total} \times t} \qquad (5)$$

where t is the chronoamperometry time, then

$$j_{HCOO^-} = \frac{FE_{HCOO^-} \times i_{total}}{\text{electrode area}} \qquad (6)$$

### In-situ Raman spectra

In situ Raman cell was custom-made from GaossUnion, China. The carbon paper (Toray H90) loaded with catalyst was set in the bottom of the Raman cell as a working electrode. The reference electrode (Ag/AgCl) and a counter electrode (Pt wire) were purchased from GaossUnion, China. Before the experiments, the electrolyte (0.5 M KHCO$_3$) was saturated with $CO_2$ (80 mL min$^{-1}$, 30 min) at room temperature and ambient pressure. A laser confocal micro spectrometer (Renishaw inVia) with a 532 nm wavelength laser was used to obtain Raman spectroscopy. For the Bi-based catalyst, the scan range of the Raman shift was set at 50–500 cm$^{-1}$.

### $^{19}$F-NMR

After a certain time of chronoamperometry test (CO$_2$ saturated 0.5 M KHCO$_3$, −0.97 V, glass carbon electrode), the catalyst was collected and used to quantify the fluorine content by $^{19}$F-NMR spectra. The fluorine content of the catalyst was quantified by $^{19}$F-NMR spectra. 150 mg samples (Bi$_2$O$_3$, Bi$_2$O$_3$-F) were dissolved by 300 μL D$_2$O and 300 μL HCl (30 wt%). After ultrasonication, the solution containing fluorine was collected by centrifugation for 10 mins (speed: 3300 g). Typical $^{19}$F-NMR sample was mixed with 500 μL solution from centrifugation and 2 μL hexafluorobenzene (internal standard).

### XPS characterization

For the ex-situ XPS characterization, the glassy carbon electrodes were used to investigate the dopant content variations during the electrochemical reactions[59,60]. To keep the original surface of the catalyst, the samples that underwent the electrochemical reactions were quickly held in a vacuum oven at 25°C until the beginning of XPS characterization.

### DFT calculation

Density functional theory (DFT), as implemented in the plane-wave Vienna ab initio simulation package (VASP) code, was used for the theoretical calculations[61,62]. The standard projector augmented wave (PAW) method was used for characterizing interaction in valence electrons and ion core[63]. The exchange-correlation function was based on generalized gradient approximation as described by Perdew-Burke-Ernzerhof (GGA-PBE)[64]. The cut-off energy is set to 520 eV and the 1 eV width Gaussian smearing was used[65]. The first Brillouin zone integration adopted a Γ-centered Monkhorst-Pack k-point grid (resolution: 0.3 Å$^{-1}$)[66]. For the correction of the van der Waals force in the calculated structure, Grimme's DFT-D3 correction with the BJ damping was used[67]. The convergence tolerances of 10$^{-4}$ eV/atom for energy and 0.02 eV/Å for maximum force were used for the geometrical optimizations. The size of Bi slab (54 atoms) were 13.60500 Å (a-axis), 14.18690 Å (b-axis), and 21.75030 Å (c-axis) respectively, where the vacuum region in the c-axis was set to a space of 15 Å to eliminate interactions between the adjacent layers. The configuration of K-point was (3, 3, 1). The Gibbs free energy of the intermediates at zero potential is calculated by using $\Delta G = \Delta E + \Delta ZPE - T\Delta S$[42,68], where $\Delta E$ is the total energy difference. $\Delta ZPE$ and $T\Delta S$ are zero-point energy correction and entropy change at room temperature (298.15 K). The computational detail for Pourbaix diagram was referred to by the previous literature[69]. The construction of a theoretical Pourbaix diagram implies deriving G(pH, U) or ΔG(pH, U) values for the possible surface structures for a broad range of pH and U. Applying an electrochemical-thermodynamic approach, the equation was as follows:

$$\Delta G(pH,U) = \Delta G(0,0) - v(H^+)k_B T(\ln 10)pH - v(e^-)eU$$

where e is the elementary charge of an electron and U is the applied electrode potential with respect to the SHE.

## Data availability

Source data are provided with this paper.

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

## Acknowledgements

This work was supported by the Key Research and Development Program sponsored by the Ministry of Science and Technology (MOST) (2022YFA1203400 and 2022YFB4002000) and the National Natural Science Foundation of China (22072145, 22102172, 22005294, 22372155, and 21925205).

## Author contributions

W.Xu. designed the idea. C.L. performed the material synthesis, characterization, experiments, and data analysis. B.M. and Z.J. performed X-ray absorption spectra (XAS) analysis. Z.S., J.G., W.Xing performed in-situ Raman analysis. P.S. performed DFT calculations. C.L., P.S., and W.Xu wrote the original draft. All authors reviewed and edited the manuscript.

## Competing interests

The authors declare no competing interests.
