## [Peer review file · Nature Communications]

REVIEWER COMMENTS

Reviewer #1 (Remarks to the Author):

This study aimed to investigate the stability of dopants and actual active sites of heterogeneous catalysts doped with nonmetal atoms. Testing the stability of dopants by varying the working potential of the reaction revealed that dopant leaching occurs at either too high or too low potential ranges. As a result, it was found that the main active site of the reaction is the newly created defective sites generated by leaching, which were unknown as active sites in previous studies. However, some experiments testing the performance and stability of the catalyst require additional analyses with some other conditions. Furthermore, additional explanations seem necessary for the results. At this stage I do not recommend this work to be published in Nature Communications.

Comments

- This work presents a window that sets the interval in which the reduction reaction that occurs depending on the voltage varies. According to this window, there is an important difference between the doped heteroatom becoming leaching and the stable remaining, so a more detailed explanation of how the scope of this window was designated is needed.
- In the test using BP, an experiment was only conducted on an electrolyte with a pH condition close to neutral. It seems that it is good to show that the pH condition is irrelevant by conducting an experiment even on a basic electrolyte with KOH, not neutral.
- In each case where N and F are doped, the leaching will occur and then defective BP will be created commonly, however, there is no mention of the difference between the two materials after that. When a very low voltage is used, it seems that there will be no effect of the heteroatom because the heteroatom has almost been removed. Therefore, it needs to explain why the difference between the two materials occurs in the CO₂RR results.
- This paper explained that the occurrence and extent of leaching vary depending on the applied voltage. However, it seems necessary to add the reasons for conducting different reactions and conditions under each voltage window.
- On Figure 4.b investigated the difference in catalytic performance between N- and F-doped materials depending on voltage and corresponding reactions. However, the performance difference was insignificant in the range of the OER reaction. In contrast to the clear differences between the two materials observed in the CO₂RR and HER, which corresponds to the same leaching window, the results were the opposite. That seems to be why additional explanations are needed.
- In this paper, it was shown in Figure S22 that the main factor of leaching is voltage, not the reaction. However, it is noted that there is no leaching occurring in the middle range of the voltage and there is no clear correlation between the voltage and the leaching rate, as shown in Figure 4. e,f. Therefore, there seems to be a missing explanation for the part where the reaction affects leaching.
- There are too many typos. (ex) Figure 4. g dopants) And there are many errors in the color representation of figure graphs. (ex) Fig.S1 (e), Fig3 (a)) and I can't find "Supplementary Information 1.1~1.7" written in the text anywhere. Unify confusing data notation. Also, the content and flow of introduction and abstract are very similar. Shorten or modify the abstract.
- In Figure 1. h on page 5, after all, Bi₂O₃-F has lower long-term durability than Bi₂O₃, Therefore, it may be difficult to say that a material with a high FE is more stable, no matter how high it is. There should be some more specific explanation.

Reviewer #2 (Remarks to the Author):

Comments:

The authors reveal the leaching of dopants with the change of working potential, which demonstrates the catalytic performance originates from the new active sites formed in situ. This work provides new insight into the activity and stability of heteroatom-doped electrocatalysts. I find the experimental sections of this paper meaningful and interesting, but the computational section needs some revision before acceptance.

Key concerns:

1. The computational section focused on the CO₂RR activities while the experimental part emphasized the instabilities of dopants in electrochemical conditions. We thought that the authors should also confirm the leaching window and stable window from a computational perspective. For example, the authors could calculate the surface Pourbaix diagram to unfold the operando surface condition, like Ref. [ACS Energy Letter, 2020, 5, 3778-3787] or directly calculate the potential window of dopant, like Ref. [Small, 2019, 15, 1901899].
2. In the experimental part, after the F leaching, it not only brings defects but also undergoes lattice expansion. How about the effect of lattice strain on the CO₂RR on Bi?

Minor issues:

3. The charge density difference and the electron transfer should be considered for the intermediate-adsorbed Bi-based catalysts.
4. The authors should mention the size of Bi slab, and the convergence test of K- points under this slab.

Reviewer #3 (Remarks to the Author):

In this manuscript, Liu et al. investigated the stability of heterogeneous nanoelectrocatalysts doped with nonmetal atoms for redox reactions. Due to the extensive applications of such hetero-atom-doped nanoelectrocatalysts, such research is fundamentally important for the precise understanding of the real catalytic mechanism of these catalysts. Based on several heteroatom-doped nanocatalysts (Bi-F, C-F and C-N) for CO₂RR, HER, ORR and OER, significantly, the authors unveiled, when the redox working potential is too low or too high, the active sites based on these dopants actually can collapse due to the fast leaching of dopants. It means that some previously observed "remarkable catalytic activity and stability" actually originated from some unknown new active sites formed in situ, indicating an operando formation of highly efficient electrocatalysts induced by heteroatom leaching. The work provides new insight into the real role of heteroatoms doped on nanocomposites for catalysis. Given this, researchers in the field of heterogeneous nanoelectrocatalysts could be inspired by such work. So, in my opinion, this work can be considered for publication in Nature Communications after the following minor revisions:

1. The specific experimental details of the in situ Raman configurations should be complemented, including electrolyte, carbon paper, and testing parameters.
2. For the DFT calculation portion, the reason for the removal of the Bi atom in Bi(012)def-F should be mentioned. In Figure 3b, the meaning of the electron localization function calculated from the optimized geometric structures needs more explanation and comparison.
3. (i) Is the rate of removal related to the atomic species (F, N atom)? A comparison is needed here. (ii) For the ex-situ XPS characterization, specific experimental details should be mentioned.
4. The authors are suggested to supplement and rearrange the reference in the introduction part to make sure a systematic overview of the latest literature.

Response to the comments

Dear Reviewers,

First of all, many thanks for your time and insightful comments on our manuscript. In the past four weeks, based on all your comments, we have revised the manuscript by supplementing large amount of new data to deal with all your concerns. All the details about the revision can be found in the following point-by-point responses. By this way, the quality of this work has been improved lot. We sincerely appreciate all your comments for the improvement of this work.

REVIEWER COMMENTS

Reviewer #1 (Remarks to the Author):

This study aimed to investigate the stability of dopants and actual active sites of heterogeneous catalysts doped with nonmetal atoms. Testing the stability of dopants by varying the working potential of the reaction revealed that dopant leaching occurs at either too high or too low potential ranges. As a result, it was found that the main active site of the reaction is the newly created defective sites generated by leaching, which were unknown as active sites in previous studies. However, some experiments testing the performance and stability of the catalyst require additional analyses with some other conditions. Furthermore, additional explanations seem necessary for the results. At this stage I do not recommend this work to be published in Nature Communications.

Response: Dear reviewer, thanks for your insightful comments on our manuscript. Based on these comments, we have done corresponding revisions carefully by supplementing more new data. More details can be found in the following point-by-point responses.

Comments

1. This work presents a window that sets the interval in which the reduction reaction that occurs depending on the voltage varies. According to this window, there is an important difference between the doped heteroatom becoming leaching and the stable remaining, so a more detailed explanation of how the scope of this window was designated is needed.

Response: Thanks for your comment here about the scope of the potential window adopted in this work. Indeed, as you know, “there is an important difference between the doped heteroatom becoming leaching and the stable remaining”. Due to the differences among different supports or dopants, the differences among different catalytic systems (certain electrocatalysts for certain redox reactions at certain electrode potentials) could be very huge. Based on the standard electrode potentials for different half-electrochemical thermodynamic reactions (Chem. Soc. Rev. 2014, 43, 631, Table R1), at a specific potential, the thermodynamically plausible half-electrochemical redox reactions are very limited, so, a single electrochemical reaction cannot be adopted to study or understand fully the dopant stability in a wide potential range (-1 V to 1.8 V vs. RHE). With this in mind, we chose to study four most typical and important energy chemistry processes (CO₂RR, HER, ORR, and OER covering the electrode potential from -1.0 V to 1.8 V vs. RHE) to reveal the

stability of different dopants on different supports.

Based on common choices of test windows for these four typical redox reactions (Nat Commun 2023, 14, 340; Nat Commun 2022, 13, 6249; Nat. Mater. 2021, 20, 1385; Nat Commun 2023, 14, 843), the typical potential values (-0.5 V, 0 V, and 1 V vs. RHE) are selected as the dividing points **approximately** for the interval sets of potential window.

Based on your comment here, we supplemented above explanations to the revised manuscript (Page 13-15) and supplementary materials (Supplemental table 4).

Table R1. The standard electrode potentials of half-electrochemical thermodynamic reactions.

Half-electrochemical reactions	E ⁰ (V vs. SHE)	E ⁰ (V vs. RHE)	Typical electrolyte pH
$O_2+4H^++4e^-\rightarrow 2H_2O$	1.229	1.288	1
$O_2+2H_2O+4e^-\rightarrow 4OH^-$	0.401	1.168	13
$2H^++2e^-\rightarrow H_2$	0	0.059	1
$2H_2O+2e^-\rightarrow H_2+2OH^-$	-0.828	-0.061	13
$CO_2+2HCO_3^-+2e^-\rightarrow CO+H_2O+2CO_3^{2-}$	-0.716	-0.300	7
$CO_2+H_2O+2e^-\rightarrow CO+2OH^-$	-0.934	-0.521	7
$CO_2+HCO_3^-+2e^-\rightarrow HCOO^-+CO_3^{2-}$	-1.030	-0.617	7
$CO_2+H_2O+2e^-\rightarrow HCOO^-+OH^-$	-1.078	-0.665	7

2. In the test using BP for CO₂RR, an experiment was only conducted on an electrolyte with a pH condition close to neutral. It seems that it is good to show that the pH condition is irrelevant by conducting an experiment even on a basic

electrolyte with KOH, not neutral.

Response: Thanks for your constructive comment here. Indeed, in this work, we only presented the CO₂RR on BP in neutral solution. As for the CO₂RR in typical three-electrode H-cell tests, it has been known that KHCO₃ is the optimal choice of electrolyte for the CO₂RR because of its buffering capacity and its ability to act as a proton source (Nat Energy 2019, 4, 732; ACS Catal. 2022, 12, 331; Nat Energy 2022, 7, 130). In the case of catalytic performance tests in alkaline electrolytes (such as KOH), firstly CO₂ molecules can react with OH⁻ and secondly water molecules would act as proton source during the electrolysis, in such case it is difficult to keep high FE of CO₂RR, and the reaction essentially becomes a HER process (J. Am. Chem. Soc. 2020, 142, 4154). Under acidic conditions, similarly, high proton concentrations can also lead to strong HER process (ACS Catal. 2021, 11, 4936). Therefore in previous work, the CO₂RR has been mainly studied extensively under neutral rather than overly basic or acidic conditions. That's why we only took the most typical CO₂RR in neutral solution as an example to show the stability of dopants. Actually we have conducted CO₂RR experiments on BP-N and BP-F in both alkaline (KOH) and acid(HClO₄) solution. In such experiments, besides the CO₂RR process, we also observed high FE_{H₂} and the fast leaching of dopants. In this work, since the effect of a pure HER on the leaching of dopants has been studied in Ar-saturated HClO₄ solution as shown in Fig.4, to study the effect of a pure CO₂RR process on the leaching of dopants, we only presented the results obtained in neutral solution in the manuscript.

3. In each case where N and F are doped, the leaching will occur and then defective BP will be created commonly, however, there is no mention of the difference between the two materials after that. When a very low voltage is used, it seems that there will be no effect of the heteroatom because the heteroatom has almost been removed. Therefore, it needs to explain why the difference between the two materials occurs in the CO₂RR results.

Response: Thanks for your insightful comment. Based on it, we supplemented the analysis of the two typical samples after the leaching of dopants via Raman spectra (Philos. Trans. R. Soc., A. 2004, 362, 2477; Adv. Mater. 2017, 29, 1603414; Angew. Chem. Int. Ed. 2019, 58, 1163; J. Am. Chem. Soc. 2019, 141, 51, 20451) as shown in the following Fig. R1. It shows the obvious difference of defects between the two materials after the leaching of dopants. After the deep leaching of different dopants, the defect structures or densities obtained on supports could be hugely different due to the different initial doping states of the N and F on carbon BP (Nano Lett. 2019, 19, 1, 530–537; ACS Catal. 2020, 10, 19, 11127–11135). Such fact can explain the difference between the two materials occurring in the CO₂RR since it has been known that the catalytic activity is dependent on the structure of the active sites (Nat Catal 2019, 2, 198–210; J. Am. Chem. Soc. 2021, 143, 279). Based on your comment here, we supplemented the new data and explanations properly in the revised manuscript (Page 13-14) and supplementary materials (Figure S22).

Figure R1. (a) Raman spectra of fresh BP-F and BP-N. (b) Raman spectra of BP-F and BP-N after leaching (CO_2 -saturated KHCO_3 , -0.87 V, 60 min).

4. This paper explained that the occurrence and extent of leaching vary depending on the applied voltage. However, it seems necessary to add the reasons for conducting different reactions and conditions under each voltage window.

Response: Thanks for your comment. The reasons for conducting different reactions and conditions under each voltage window are the following: It has been known that heterogeneous nano-electrocatalysts doped with nonmetal atoms for all kinds of redox reactions have been studied extensively based on the so-called dopant-based active sites. These redox reactions occur selectively in different potential regions and different electrolyte solutions due to their different redox potentials as shown in above Table R1. To understand reliably the stabilities of these dopants during the catalysis process for different reactions, take the BP-N for ORR in acidic solution as an example, one should study the durability of N in O_2 -saturated acidic solution in the typical potential range between 0.0 V and 1.0 V vs. RHE. Based on your comment here, we have supplemented such reason to the revised manuscript (Page 13-14).

5. On Figure 4.b investigated the difference in catalytic performance between N- and F-doped materials depending on voltage and corresponding reactions. However, the performance difference was insignificant in the range of the OER reaction. In contrast to the clear differences between the two materials observed in the CO₂RR and HER, which corresponds to the same leaching window, the results were the opposite. That seems to be why additional explanations are needed.

Response: Thanks for your comment here. It has been well-known that the activities of same type of catalysts prepared with different methods for the same reaction could be hugely different. Also, a catalyst could present different catalytic activities for different reactions due to the different catalytic mechanisms. So, it is not surprise for us to observe the similar or different activities between BP-N and BP-F for ORR or CO₂RR. In this work, it does not matter whether the activity difference between different catalysts for the same reaction is big or not since it has been known that the catalytic activity of a catalyst depends on many factors, such as the materials adopted, the surface pore structures, the structure of the active sites, etc. (Chem. Soc. Rev., 2021, 50, 7745; Chem. Soc. Rev., 2021, 50, 11785; Adv. Mater. 2016, 28, 9532; J. Energy Chem. 2017, 26, 422; ACS Catal. 2022, 12, 1216; ACS Nano 2022, 16, 15273). Based on your comment here, we have supplemented an explanation properly in the revised manuscript (Page 13-14).

6. In this paper, it was shown in Figure S22 that the main factor of leaching is voltage, not the reaction. However, it is noted that there is no leaching occurring in the middle range of the voltage and there is no clear correlation between the voltage and the

leaching rate, as shown in Figure 4. e,f. Therefore, there seems to be a missing explanation for the part where the reaction affects leaching.

Response: Thanks for your comment here. It has been known that the stability of dopants on supports is dependent on the electrode potential (Nano Lett. 2019, 19, 1, 530–537, ACS Catal. 2020, 10, 19, 11127–11135). The too low or too high voltage can hugely affect or damage the surface structure of a catalyst. Based on your reminder, we supplemented the calculation of leaching rates at different potentials as shown in the following Fig. R2. It shows clearly the correlation between the voltage and the leaching rate. Based on your comment here, we supplemented such new data and explanation to the revised manuscript (Page 13-14 and Figure 4f, h).

Figure R2. (a) F-dopant and (b) N-dopant leaching rate as a function of potential applied.

7. There are too many typos. (ex) Figure 4. g dopants) And there are many errors in the color representation of figure graphs. (ex) Fig.S1 (e), Fig3 (a)) and I can't find "Supplementary Information 1.1~1.7" written in the text anywhere. Unify confusing data notation. Also, the content and flow of introduction and abstract are very similar. Shorten or modify the abstract.

Response: Thanks for your reminder. We are very sorry for the errors left in the manuscript. Based on your reminder, we have double-checked the manuscript and made the corresponding corrections and improved the language, including all the points you mentioned.

8. In Figure 1. h on page 5, after all, $\text{Bi}_2\text{O}_3\text{-F}$ has lower long-term durability than Bi_2O_3 , Therefore, it may be difficult to say that a material with a high FE is more stable, no matter how high it is. There should be some more specific explanation.

Response: Thanks for your comment here. But it looks like that you misunderstood the data shown in Figure 1h. Actually, the data shown in it or in the following Figure R3 indicate firmly that $\text{Bi}_2\text{O}_3\text{-F}$ has higher rather than lower long-term durability than Bi_2O_3 : It shows clearly, after a long-term CO_2RR process, the FE on $\text{Bi}_2\text{O}_3\text{-F}$ only decreased 11%, much lower than the FE decrease of 22% on Bi_2O_3 . We made such judgment based on the decreased percentage of FE after a long-term catalytic process (Deactivation of Catalysts, Academic Press, London, 1984). Based on your reminder here, to avoid confusing readers, we supplemented detailed explanations in the revised manuscript (Page 6). We thank you for your reminder here.

Figure R3. Long-term durability of formate selectivity of the Bi₂O₃ (blue), and Bi₂O₃-F (red) under chronoamperometry test (-0.97 V vs. RHE, CO₂-saturated 0.5 M KHCO₃).

Reviewer #2 (Remarks to the Author):

Comments:

The authors reveal the leaching of dopants with the change of working potential, which demonstrates the catalytic performance originates from the new active sites formed in situ. This work provides new insight into the activity and stability of heteroatom-doped electrocatalysts. I find the experimental sections of this paper meaningful and interesting, but the computational section needs some revision before acceptance.

Response: Dear reviewer, we thank you for your insightful comments on our work. Based on these comments, we supplemented new DFT calculations to further improve the quality of this work. More details can be found in the following point-by-point responses to your comments.

Key concerns:

1. The computational section focused on the CO₂RR activities while the experimental part emphasized the instabilities of dopants in electrochemical conditions. We thought that the authors should also confirm the leaching window and stable window from a computational perspective. For example, the authors could calculate the surface Pourbaix diagram to unfold the operando surface condition, like Ref. [ACS Energy Letter, 2020, 5, 3778-3787] or directly calculate the potential window of dopant, like Ref. [Small, 2019, 15, 1901899].

Response: Thanks for your constructive comments. In this revision, based on your comment here, we supplemented the calculation about the Pourbaix diagram to

unfold the operando surface condition of C-F and C-N, like Ref. [ACS Energy Letter, 2020, 5, 3778-3787]. As shown in the following Figure R4, starting from the dopant stable form in different pH and potential ranges, we calculated the potential-pH relationship associated with the dopant leaching process. The obtained leaching Pourbaix diagram reveals that the F- and N-dopants leached at either too high or too low potential ranges while a relatively stable potential interval existed in the middle range of the potential, which was in good agreement with the experimental outcomes (the relative stabilities of heteroatoms shown in Fig. 4).

Based on your comment here, we have supplemented these new data in the revised manuscript (Page 14-15) and SI (Figure S27). The calculation details were supplemented to **Methods** part (Page-20).

Figure R4. Pourbaix diagram of (a) F- and (b) N-dopant on carbon.

2. In the experimental part, after the F leaching, it not only brings defects but also undergoes lattice expansion. How about the effect of lattice strain on the CO_2RR on Bi?

Response: Thanks for your insightful comment here. Based on it, we supplemented

the calculation about the effect of lattice strain on CO₂RR on Bi (ACS Catal. 2021, 11, 6662). In this work, we indeed observed that the Bi_{def-F} undergoes lattice expansion after the F leaching (XRD spectra shown in Fig. 2f). The XRD refinement results of Bi_{def-F} show a 1% lattice expansion along the a-axis (Table R2). Based on such fact, in the modeling for the DFT calculations, the lattice of Bi(012) was expanded by 1% along the a-axis to investigate the possible effect of lattice strain on the CO₂RR (denoted as Bi(012)_{1%-expansion})(Figure Ra-c). Firstly, the new calculation results of charge density difference and bader charge (Figure Rd) show no significant change in the electron transfer of intermediate-adsorbed Bi(012)_{1% expansion} surface as compared to the Bi(012) surface without expansion. While the free energy profiles revealed that the rate-determining step of CO₂RR on Bi(012)_{1% expansion} is the CO₂ activation step, hugely different from that on Bi(012) (Figure R5e,f). Such fact confirms the point you mentioned about the strain effect on the CO₂RR on Bi.

Based on your reminder here, we supplemented the new result to the revised manuscript (Page-11) and SI (Supplemental table 3, Fig. S16, Supplemental note 1).

Table R2. The XRD refinement results of $\text{Bi}_2\text{O}_3\text{-F}$, Bi-F , and $\text{Bi}_{\text{def-F}}$.

Parameters	$\text{Bi}_2\text{O}_3\text{-F}$	Bi-F	$\text{Bi}_{\text{def-F}}$
a (Å)	7.73241	4.53488	4.58843
b (Å)	7.73241	4.53488	4.58843
c (Å)	5.62584	11.81400	11.81400
α (°)	90.0000	90.0000	90.0000
β (°)	90.0000	90.0000	90.0000
γ (°)	90.0000	120.0000	120.0000
V (Å ³)	336.370	210.406	215.405
R_{wp} (%)	3.853	3.517	2.258
χ^2	2.90	6.17	2.44

Figure R5. (a) The lattice of $\text{Bi}(012)$ was expanded by 1% along the a-axis to investigate the effect of lattice strain on the CO_2RR (denoted as $\text{Bi}(012)1\%$ -expansion). (b) Optimized adsorption structures of $^*\text{OCHO}$ for $\text{Bi}(012)1\%$ -expansion surface. (c) The charge density difference in intermediate-adsorbed $\text{Bi}(012)1\%$ -expansion surface. The yellow color represents charge

accumulation and blue represents charge dissipation in space. (d) Bader charge of active sites and (e) binding energy of *OCHO on the different catalytic surface (Bi(012), Bi(012)_{1%-expansion}). (f) Free energy profiles for the formation of *OCHO intermediate on the catalytic surface.

Minor issues:

3. The charge density difference and the electron transfer should be considered for the intermediate-adsorbed Bi-based catalysts.

Response: Thanks for your suggestion. Based on it, we supplemented the calculation about the charge density difference and the electron transfer (bader charge) in each intermediate (*OCHO)-adsorbed Bi-based catalyst (Figure R). It can be seen that the F doping can weaken the electron transfer from Bi to O and then weaken the adsorption of the intermediate on Bi, while after the formation of defects via the leaching of F, the electron transfer from Bi to O can be enhanced, which then enhances the adsorption of the intermediate on Bi.

Based on your comment here, we supplemented these new data to the revised manuscript (Page 10-11) and SI (Figure S15).

Figure R6. The charge density difference and the electron transfer (bader charge) in each intermediate-adsorbed Bi-based catalyst. The yellow color represents charge accumulation and blue represents charge dissipation in space.

4. The authors should mention the size of Bi slab, and the convergence test of K-points under this slab.

Response: Thanks for your suggestion here. For this case, the size of Bi slab (54 atoms) were 13.60500 Å (a-axis), 14.18690 Å (b-axis), and 21.75030 Å (c-axis) respectively, where the vacuum region in the c-axis was set to a space of 15 Å to eliminate interactions between the adjacent layers. The convergence test results of total-energy calculations with respect to the number of K-points are shown in Figure R. We chose the configuration of K-point (3, 3, 1) for our calculation here. Based on your comment here, we have supplemented these details in the **Methods** part of the revised manuscript (Page 19).

Figure R7. The convergence test of Bi slab of total-energy calculations with respect to the number of K-points.

Reviewer #3 (Remarks to the Author):

In this manuscript, Liu et al. investigated the stability of heterogeneous nanoelectrocatalysts doped with nonmetal atoms for redox reactions. Due to the extensive applications of such hetero-atom-doped nanoelectrocatalysts, such research is fundamentally important for the precise understanding to the real catalytic mechanism of these catalysts. Based on several heteroatom-doped nanocatalysts (Bi-F, C-F and C-N) for CO₂RR, HER, ORR and OER, significantly, the authors unveiled, when the redox working potential is too low or too high, the active sites based on these dopants actually can collapse due to the fast leaching of dopants. It means that some previously observed “remarkable catalytic the activity and stability” actually originated from some unknown new active sites formed in situ, indicating an operando formation of highly efficient electrocatalysts induced by heteroatom leaching. The work provides new insight into the real role of heteroatoms doped on nanocomposites for catalysis. Given this, researchers in the field of heterogeneous nanoelectrocatalysts could be inspired by such work. So, in my opinion, this work can be considered for publication in Nature Communications after the following minor revisions:

Response: We are very grateful for your comments on our work. Based on these comments, we have done corresponding revisions carefully by supplementing more new data. More details can be found in the following point-by-point responses.

1. The specific experimental details of the in situ Raman configurations should be complemented, including electrolyte, carbon paper, and testing parameters.

Response: Thanks for your comment here. Based on your comment here, we supplemented all the details about the in situ Raman tests to the **Methods** Part in revised manuscript (Page-19).

2. For the DFT calculation portion, the reason for the removal of the Bi atom in $\text{Bi}(012)_{\text{def-F}}$ should be mentioned. In Figure 3b, the meaning of the electron localization function calculated from the optimized geometric structures needs more explanation.

Response: Thanks for your comments here. The reason for the removal of the Bi atom in the DFT calculation is based on the experimental observation about the structural evolution of $\text{Bi}_2\text{O}_3\text{-F}$ during the CO_2RR process (Figure 2). It shows clearly that the leaching of F atoms from the Bi-F surface can induce the local dislocation and then the in-situ formation of defective sites. Since the defects are also catalytically active for CO_2RR (Nat Commun. 2019, 10, 2807; J. Am. Chem. Soc. 2020, 142, 13, 6400), then, in the DFT calculation, we constructed defective $\text{Bi}(012)$ to elucidate the catalytic activity of the leaching-induced Bi defects for CO_2RR .

As for the electron localization function (ELF), it is a measure of the likelihood of finding an electron in the neighborhood space with the same spin (J. Chem. Phys. 1990, 92 (9), 5397; Angew. Chem. Int. Ed. Engl. 1997, 36, 1808). The ELF has a value between 0 and 1, with an upper value of 1 indicating complete localization of the electron, while a value of 0 may indicate complete delocalization of the electron. The $\text{Bi}(012)_{\text{def-F}}$ displays higher electron delocalization around the formed defective sites compared with the $\text{Bi}(012)$ and $\text{Bi}(012)\text{-F}$. We have added the explanations in

the revised manuscript (Page-10).

3. (i) Is the rate of removal related to the atomic species (F, N atom)? A comparison is needed here. (ii) For the ex-situ XPS characterization, specific experimental details should be mentioned.

Response: Thanks for your insightful comment here. Based on it, we supplemented the calculation about the potential-dependent leaching rates of N and F from the data shown in Fig. 4 or in the following Figure R8. It shows clearly that the potential-dependent leaching rate of N is different from that of F probably due to the different bindings of N and F on carbon (Adv. Mater., 2013, 25, 6879-6883, Nat Commun 2017, 8, 15938). More details can be found on Page-14-15 in the revised manuscript.

Figure R8. Comparisons of leaching rates between the different types of dopants (F-dopants and N-dopants).

As for specific experimental details for the ex-situ XPS characterization, based on your reminder, based on your comment here, we supplemented all these details to the **Methods** part in revised manuscript (Page-19).

4. The authors are suggested to supplement and rearrange the reference in the

introduction part to make sure a systematic overview of the latest literature.

Response: Thanks for your comment here. Based on it, we reorganized the references in the introduction part to make sure a systematic overview of the latest literature by supplementing the citation of some new relevant literatures (for example: Nat Catal. 2020, 3, 478; Nat Energy 2020, 5, 478; ACS Catal. 2021, 11, 12, 7604; Nat Commun. 2022, 13, 2205). We have marked the key revisions by a yellow background in the revised reference.

REVIEWER COMMENTS

Reviewer #2 (Remarks to the Author):

No further comment.

Reviewer #3 (Remarks to the Author):

Authors have addressed all concerns and the quality of paper has been improved. Its publication is recommended.

Reviewer #5 (Remarks to the Author):

Remarks to the author:

We primarily assessed the adequacy of the authors' responses to Reviewer 1's comments and provided detailed additional comments on each question and answer below. However, it is worth noting that some of the questions have not been sufficiently addressed, necessitating further improvement. Furthermore, when considering the study from an overall perspective, it's important to acknowledge that the key concept the authors are attempting to establish—namely, the leaching of dopants at both high and low potentials and the subsequent formation of new active sites—may not apply universally to all electrolytes or host materials. For instance, it is well-documented that chloride containing electrolytes, such as HClO₄ used in this study, significantly influence the leaching of fluorine dopants (Sci. Rep. 2017, 7, 4595; J. Electrochem. Soc. 1999, 146, 977; J. Electrochem. Soc. 1996, 143, 442). In contrast, electrolytes like H₂SO₄ do not significantly impact the leaching of fluorine dopants, even at high potentials (Nat Commun. 2022, 13, 2668; Nano Energy, 2019, 65, 104008). Similarly, even at low potentials, fluorine dopants in tin oxide show good performance and durability in KOH and KHCO₃ electrolytes during CO₂RR without dopant leaching issues (Nat Commun. 2022, 13, 2205). We believe that these cases should also be fully considered in this study.

Comment 1

The range of potential windows to address the four reactions presented in this study is well thought out, and we think the authors' responses are appropriate.

Comment 2

The reviewer pointed out that the CO₂RR test of BP samples was evaluated in a neutral electrolyte and asked if the results were the same for non-neutral electrolytes. In response, the authors said that for the CO₂RR test, only neutral electrolytes were considered, as it is generally advantageous to perform the test in neutral, but that the CO₂RR was also evaluated in KOH and HClO₄ for both BP-N and BP-F. However, according to our review, the CO₂RR experimental results in KOH and HClO₄ for BP-N and BP-F have not been updated, and this needs to be corrected.

Comment 3

The reviewer pointed out the cause of the performance difference between the two samples after the dopants of N and F were sufficiently leached in CO₂RR, and the authors explained the performance difference by the difference in defect structure through Raman analysis results. However, the Raman analysis results after leaching showed that the DG ratio increased similarly for BP-F and BP-N, but the performance difference of CO₂RR is very large between them. It seems that the authors' answer is not enough to explain the cause of the performance difference between the two samples, and it needs to be further supplemented.

Comment 4

The authors have responded appropriately to reviewer's comment.

Comment 5

The reviewer pointed out that BP-N and BP-F do not perform differently in the OER region, unlike CO₂RR and HER and the authors explained that the same type of catalyst made in different ways may have different activities and may show different catalytic activities for different reactions. However, the authors' answer is somewhat vague at this point, and we believe it needs to be more specific.

Comment 6

Appropriate new data have been presented to address the points made by the reviewer. However, as mentioned earlier, it is important to note that the authors' claim that too low or too high a potential can significantly affect or damage the catalyst surface may depend on the type of electrolyte or the type of host material of the dopant.

Comment 7

The authors have responded appropriately to reviewer's comment.

Comment 8

The authors have responded appropriately to reviewer's comment.

Response to the comments

Dear Reviewers,

First of all, many thanks for your comments on our work. In the past two more months, based on all your comments, we have supplemented many new data to deal with all your concerns and provide a point-by-point response to these comments. With all these revisions made based on your comments, the quality of this work has been improved lot. We thank you all very much for your time and work on our manuscript.

All the details about the revision can be found in the following contents or in the revised manuscript.

Best regards.

Weilin Xu

REVIEWER COMMENTS

Reviewer #2 (Remarks to the Author):

No further comment.

Response: We thank the reviewer for his or her previous insightful comments which helped lot for the improvement of this work.

Reviewer #3 (Remarks to the Author):

Authors have addressed all concerns and the quality of paper has been improved. Its publication is recommended.

Response: We thank the reviewer for the positive comments and recommendation on the publication of this work, also thank his or her previous insightful comments which helped lot for the improvement of this work.

Reviewer #5 (Remarks to the Author):

We primarily assessed the adequacy of the authors' responses to Reviewer 1's comments and provided detailed additional comments on each question and answer below. However, it is worth noting that some of the questions have not been sufficiently addressed, necessitating further improvement. Furthermore, when considering the study from an overall perspective, it's important to acknowledge that the key concept the authors are attempting to establish—namely, the leaching of dopants at both high and low potentials and the subsequent formation of new active sites—may not apply universally to all electrolytes or host materials. For instance, it is well-documented that chloride containing electrolytes, such as HClO_4 used in this study, significantly influence the leaching of fluorine dopants (Sci. Rep. 2017, 7, 4595; J. Electrochem. Soc. 1999, 146, 977; J. Electrochem. Soc. 1996, 143, 442). In contrast, electrolytes like H_2SO_4 do not significantly impact the leaching of fluorine dopants, even at high potentials (Nat Commun. 2022, 13, 2668; Nano Energy, 2019, 65, 104008). Similarly, even at low potentials, fluorine dopants in tin oxide show good performance and durability in KOH and KHCO_3 electrolytes during CO_2RR without dopant leaching issues (Nat Commun. 2022, 13, 2205). We believe that these cases should also be fully considered in this study.

Response: Dear reviewer, we thank you for your insightful comments on our work. Based on these additional comments, we have done corresponding revisions carefully by supplementing more new data. More details about the response to your above concerns can be found in the following point-by-point responses.

Comments

1. The range of potential windows to address the four reactions presented in this study is well thought out, and we think the authors' responses are appropriate.

Response: Thanks for your comment.

2. The reviewer pointed out that the CO₂RR test of BP samples was evaluated in a neutral electrolyte and asked if the results were the same for non-neutral electrolytes. In response, the authors said that for the CO₂RR test, only neutral electrolytes were considered, as it is generally advantageous to perform the test in neutral, but that the CO₂RR was also evaluated in KOH and HClO₄ for both BP-N and BP-F. However, according to our review, the CO₂RR experimental results in KOH and HClO₄ for BP-N and BP-F have not been updated, and this needs to be corrected.

Response: Thanks for your comment here. In our response to the Reviewer #1, we mentioned the poor CO₂RR performance of some reported catalysts (J. Electroanal. Chem., 190 (1985) 157-170; J. Am. Chem. Soc. 2023, 145, 6762; J. Am. Chem. Soc. 2020, 142, 4154) in overly basic or acidic conditions, so we didn't put that part of the data into our paper. In the present revision, based on your comment here, we supplement new data about the CO₂RR on BP-N and BP-F in both KOH and HClO₄, respectively, as shown in the following Figure R1. It shows clearly that the CO₂RR activity of both BP-N and BP-F in KOH or HClO₄ is negligible compared with that shown in neutral electrolyte shown in Fig. R1g or **Supplementary Fig. 20 in the revised manuscript**. Moreover, Fig. R1 also shows the fast leaching of N and F even in KOH and HClO₄, approximately the same as that in KHCO₃ solution (Fig. 4e,g), confirming that the electrode potential is the main reason for the dopant leaching.

Figure R1. (a)-(d) LSV and corresponding FE_{CO} at different applied potentials of BP-F and BP-N in 0.1 M HClO₄ and 0.1 M KOH. Colours in red and blue represent CO₂ and Ar saturated electrolytes respectively. (e) F-dopant and (f) N-dopant content as a function of time of the potential (-0.87 V) applied in 0.1 M HClO₄ and 0.1 M KOH. (g) Comparison of FE_{CO} at different applied potentials ranging from -0.57 V to -1.07 V (RHE) in CO₂ saturated 0.5 M KHCO₃.

3. The reviewer pointed out the cause of the performance difference between the two samples after the dopants of N and F were sufficiently leached in CO₂RR, and the authors explained the performance difference by the difference in defect structure through Raman analysis results. However, the Raman analysis results after leaching showed that the DG ratio increased similarly for BP-F and BP-N, but the performance difference of CO₂RR is very large between them. It seems that the authors' answer is not enough to explain the cause of the performance difference between the two samples, and it needs to be further supplemented. **5.** The reviewer

pointed out that BP-N and BP-F do not perform differently in the OER region, unlike CO₂RR and HER and the authors explained that the same type of catalyst made in different ways may have different activities and may show different catalytic activities for different reactions. However, the authors' answer is somewhat vague at this point, and we believe it needs to be more specific.

Response: Thanks for your insightful **comment-3** and **-5** shown above about the activity difference between BP-N and BP-F for different reactions. The DG ratio revealed from Raman analysis (Supplementary Fig. 22) suggests higher degree of defects on BP-N (Adv. Mater. 2017, 29, 1603414; Angew. Chem. Int. Ed. 2019, 58, 1163; J. Am. Chem. Soc. 2019, 141, 51, 20451). Moreover, the structures of the defects obtained after the leaching of F and N may be also different from each other since the defects could be in all kinds of structures (Adv. Funct. Mater. 2020, 30, 2001097; Applied Catalysis B: Environmental 2021, 295, 120291), then the defects in different structures could present different activities for different or even the same reactions. Here, based on your new comments here, to further explain the cause of the performance difference between the two samples, we further supplemented the measurements of electrochemical double layer capacitance (C_{dl}) of BP-N and BP-F in the 0.5M KHCO₃ before and after the leaching of dopants. As shown in the following Figure R2, the C_{dl} values of BP-F before and after the leaching is hugely different from that of BP-N, indicating the difference in the electrochemical active surface area (ECSA) between BP-F and BP-N. Such difference could induce hugely different activity and selectivity of CO₂RR (J. Am. Chem. Soc. 2021, 143, 1, 279).

Based on your comments here, we supplemented the new data and explanations

properly in the revised manuscript (Page 8) and supplementary materials (Supplementary Fig. 23).

On the other hand, in this work, it does not matter whether the activity difference between different catalysts is big or small. The key concept we attempted to establish is the leaching of dopants at high or low potentials and the subsequent formation of new active sites. As for the detailed reason for the performance difference between the two samples for the same or different reaction, based on your reminder, we will make a deep study in future to make it clear systemically.

Figure R2. (a)-(c) CV profiles of BP-F and BP-N in the non-Faradaic region of 0 – 0.1 V vs. RHE with the scan rate of 10, 20, 30, 40, 50 mV s^{-1} and corresponding C_{dl} plots obtained from the CV curves. (d)-(f) The C_{dl} test results after the dopant leaching (CO_2 -saturated KHCO_3 , -0.87 V, 60 min).

4. The authors have responded appropriately to reviewer's comment.

Response: Thanks for your comment.

6. Appropriate new data have been presented to address the points made by the reviewer. However, as mentioned earlier, it is important to note that the authors' claim that too low or too high a potential can significantly affect or damage the catalyst surface may depend on the type of electrolyte or the type of host material of the dopant. (For instance, it is well-documented that chloride containing electrolytes, such as HClO₄ used in this study, significantly influence the leaching of fluorine dopants (Sci. Rep. 2017, 7, 4595; J. Electrochem. Soc. 1999, 146, 977; J. Electrochem. Soc. 1996, 143, 442). In contrast, electrolytes like H₂SO₄ do not significantly impact the leaching of fluorine dopants, even at high potentials (Nat Commun. 2022, 13, 2668; Nano Energy, 2019, 65, 104008). Similarly, even at low potentials, fluorine dopants in tin oxide show good performance and durability in KOH and KHCO₃ electrolytes during CO₂RR without dopant leaching issues (Nat Commun. 2022, 13, 2205). We believe that these cases should also be fully considered in this study.)

Response: Thanks for your insightful comment here. Based on it, for the effect of different types of electrolytes, we supplemented new experiments in 0.05 M H₂SO₄ for HER, ORR, and OER on these catalysts. As shown in the following Figure R3, R4, the potential-dependent leaching of dopants (N, F) observed in H₂SO₄ electrolyte is the same as that observed in 0.1 M HClO₄, further confirming the potential-dependent leaching of dopants, and the types of electrolytes do not affect much the leaching of dopants.

About the references you mentioned above (Nat Commun. 2022, 13, 2668; Nano Energy, 2019, 65, 104008; Nat Commun. 2022, 13, 2205), we read these references carefully, and found the authors in these references actually only observed the remarkable durability of performance, but without directly analyzing the possible

leaching of dopants. Here, based on your comment here, to make it clear about the stability of dopants during the electrocatalysis, we took the fluorinated tin oxide ($\text{SnO}_2\text{-F}$) for CO_2RR you mentioned (Nat Commun. 2022, 13, 2205) as an example, to make it clear about the stability of F on SnO_2 . As shown in Figure R5 a-c, the $\text{SnO}_2\text{-F}$ presents remarkable performance stability, consistent with the observation presented in literature (Nat Commun. 2022, 13, 2205). To further reveal the stability of F on SnO_2 in this case, the fluoride species on $\text{SnO}_2\text{-F}$ were investigated with X-ray photoelectron spectra after certain time of CO_2RR at -0.97 V. As shown in Figure R5d, after tens of minute of CO_2RR , the fluorine content on $\text{SnO}_2\text{-F}$ dropped seriously, and then fluorine almost disappeared after 90 minutes. Such fact indicates that the observed remarkable durability of performance of F-SnO_2 for CO_2RR (Nat Commun. 2022, 13, 2205) is not from the F on Sn, but from the new active sites on Sn formed after the leaching of F, consistent with the conclusion made in this work.

Based on your comment here, we supplemented the new data and explanations properly in the revised manuscript (Page 8-10) and supplementary materials (Supplementary Fig. 28-30).

Figure R3. (a-c) The linear sweep curves of the catalyst (rotation rate 1600 rpm, scan rate of 5 mV/s). (a) HER was tested in Ar-saturated 0.05 M H₂SO₄, (b) ORR and (c) OER were tested in O₂-saturated 0.05 M H₂SO₄. Colours in black, red and blue represent BP, BP-F, and BP-N, respectively. (d-f) F-dopant and (g-i) N-dopant content as a function of potential applied time in the 0.05 M H₂SO₄, (d, g) HER, (e, h) ORR, and (f, i) OER.

Figure R4. Comparisons of (a-c) F-dopant and (d-f) N-dopant content variations in the different electrolyte at same applied potential.

Figure R5. (a) LSV of SnO₂-F with pH corrections for CO₂ and Ar saturated electrolytes. (b) FE_{HCOO-} of SnO₂-F at different applied potentials ranging from -0.77 V to -1.17 V (RHE). (c) Long-term durability of formate selectivity of the SnO₂-F under chronoamperometry test (0.5 M KHCO₃). (d) Time-dependent XPS of SnO₂-F after CO₂RR at -0.97 V.

7.The authors have responded appropriately to reviewer's comment.

Response: Thanks for your comment here.

8. The authors have responded appropriately to reviewer's comment.

Response: Thanks for your comment here.

REVIEWERS' COMMENTS

Reviewer #5 (Remarks to the Author):

The authors have fully addressed all comments and we believe that the quality of the paper is much improved for publication.